# Predictive features of ligand-specific signaling through the estrogen receptor

Jerome C Nwachukwu[1,†], Sathish Srinivasan[1,†], Yangfan Zheng[2], Song Wang[2], Jian Min[3], Chune Dong[2], Zongquan Liao[2], Jason Nowak[1], Nicholas J Wright[1], René Houtman[4], Kathryn E Carlson[3], Jatinder S Josan[5], Olivier Elemento[6], John A Katzenellenbogen[3,**], Hai-Bing Zhou[2,***] & Kendall W Nettles[1,*]

## Abstract

Some estrogen receptor-α (ERα)-targeted breast cancer therapies such as tamoxifen have tissue-selective or cell-specific activities, while others have similar activities in different cell types. To identify biophysical determinants of cell-specific signaling and breast cancer cell proliferation, we synthesized 241 ERα ligands based on 19 chemical scaffolds, and compared ligand response using quantitative bioassays for canonical ERα activities and X-ray crystallography. Ligands that regulate the dynamics and stability of the coactivator-binding site in the C-terminal ligand-binding domain, called activation function-2 (AF-2), showed similar activity profiles in different cell types. Such ligands induced breast cancer cell proliferation in a manner that was predicted by the canonical recruitment of the coactivators NCOA1/2/3 and induction of the *GREB1* proliferative gene. For some ligand series, a single inter-atomic distance in the ligand-binding domain predicted their proliferative effects. In contrast, the N-terminal coactivator-binding site, activation function-1 (AF-1), determined cell-specific signaling induced by ligands that used alternate mechanisms to control cell proliferation. Thus, incorporating systems structural analyses with quantitative chemical biology reveals how ligands can achieve distinct allosteric signaling outcomes through ERα.

**Keywords** Breast cancer; Chemical biology; Crystal structure; Nuclear receptor; Signal transduction

**Subject Categories** Chemical Biology; Structural Biology; Transcription

**Mol Syst Biol. (2016) 12: 864**

## Introduction

Many drugs are small-molecule ligands of allosteric signaling proteins, including G protein-coupled receptors (GPCRs) and nuclear receptors such as ERα. These receptors regulate distinct phenotypic outcomes (*i.e.,* observable characteristics of cells and tissues, such as cell proliferation or the inflammatory response) in a ligand-dependent manner. Small-molecule ligands control receptor activity by modulating recruitment of effector enzymes to distal regions of the receptor, relative to the ligand-binding site. Some of these ligands achieve selectivity for a subset of tissue- or pathway-specific signaling outcomes, which is called selective modulation, functional selectivity, or biased signaling, through structural mechanisms that are poorly understood (Frolik *et al*, 1996; Nettles & Greene, 2005; Overington *et al*, 2006; Katritch *et al*, 2012; Wisler *et al*, 2014). For example, selective estrogen receptor modulators (SERMs) such as tamoxifen (Nolvadex®; AstraZeneca) or raloxifene (Evista®; Eli Lilly) (Fig 1A) block the ERα-mediated proliferative effects of the native estrogen, 17β-estradiol (E2), on breast cancer cells, but promote beneficial estrogenic effects on bone mineral density and adverse estrogenic effects such as uterine proliferation, fatty liver, or stroke (Frolik *et al*, 1996; Fisher *et al*, 1998; McDonnell *et al*, 2002; Jordan, 2003).

ERα contains structurally conserved globular domains of the nuclear receptor superfamily, including a DNA-binding domain (DBD) that is connected by a flexible hinge region to the ligand-binding domain (LBD), as well as unstructured AB and F domains at its amino and carboxyl termini, respectively (Fig 1B). The LBD contains a ligand-dependent coactivator-binding site called activation function-2 (AF-2). However, the agonist activity of SERMs derives from activation function-1 (AF-1)—a coactivator recruitment site located in the AB domain (Berry *et al*, 1990; Shang & Brown, 2002; Abot *et al*, 2013).

1　Department of Cancer Biology, The Scripps Research Institute, Jupiter, FL, USA
2　State Key Laboratory of Virology, Key Laboratory of Combinatorial Biosynthesis and Drug Discovery (Wuhan University), Ministry of Education, Wuhan University School of Pharmaceutical Sciences, Wuhan, China
3　Department of Chemistry, University of Illinois, Urbana, IL, USA
4　PamGene International, Den Bosch, The Netherlands
5　Department of Chemistry, Virginia Tech, Blacksburg, VA, USA
6　Department of Physiology and Biophysics, Institute for Computational Biomedicine, Weill Cornell Medical College, New York, NY, USA
　*Corresponding author. Tel: +1 561 228 3209; E-mail: knettles@scripps.edu
　**Corresponding author. Tel: +1 217 333 6310; E-mail: jkatzene@illinois.edu
　***Corresponding author. Tel: +86 27 68759586; E-mail: zhouhb@whu.edu.cn
　†These authors contributed equally to this work

**Figure 1. Allosteric control of ERα activity.**

A   Chemical structures of some common ERα ligands. BSC, basic side chain. E2-rings are numbered A-D. The E-ring is the common site of attachment for BSC found in many SERMS.
B   ERα domain organization lettered, A-F. DBD, DNA-binding domain; LBD, ligand-binding domain; AF, activation function
C   Schematic illustration of the canonical ERα signaling pathway.
D   Linear causality model for ERα-mediated cell proliferation.
E   Branched causality model for ERα-mediated cell proliferation.

AF-1 and AF-2 bind distinct but overlapping sets of coregulators (Webb *et al*, 1998; Endoh *et al*, 1999; Delage-Mourroux *et al*, 2000; Yi *et al*, 2015). AF-2 binds the signature LxxLL motif peptides of coactivators such as NCOA1/2/3 (also known as SRC-1/2/3). AF-1 binds a separate surface on these coactivators (Webb *et al*, 1998; Yi *et al*, 2015). Yet, it is unknown how different ERα ligands control AF-1 through the LBD, and whether this inter-domain communication is required for cell-specific signaling or anti-proliferative responses.

In the canonical model of the ERα signaling pathway (Fig 1C), E2-bound ERα forms a homodimer that binds DNA at estrogen-response elements (EREs), recruits NCOA1/2/3 (Metivier *et al*,

2003; Johnson & O'Malley, 2012), and activates the *GREB1* gene, which is required for proliferation of ERα-positive breast cancer cells (Ghosh *et al*, 2000; Rae *et al*, 2005; Deschenes *et al*, 2007; Liu *et al*, 2012; Srinivasan *et al*, 2013). However, ERα-mediated proliferative responses vary in a ligand-dependent manner (Srinivasan *et al*, 2013); thus, it is not known whether this canonical model is widely applicable across diverse ERα ligands.

Our long-term goal is to be able to predict proliferative or anti-proliferative activity of a ligand in different tissues from its crystal structure by identifying different structural perturbations that lead to specific signaling outcomes. The simplest response model for ligand-specific proliferative effects is a linear causality model, where

the degree of NCOA1/2/3 recruitment determines *GREB1* expression, which in turn drives ligand-specific cell proliferation (Fig 1D). Alternatively, a more complicated branched causality model could explain ligand-specific proliferative responses (Fig 1E). In this signaling model, multiple coregulator binding events and target genes (Won Jeong *et al*, 2012; Nwachukwu *et al*, 2014), LBD conformation, nucleocytoplasmic shuttling, the occupancy and dynamics of DNA binding, and other biophysical features could contribute independently to cell proliferation (Lickwar *et al*, 2012).

To test these signaling models, we profiled a diverse library of ERα ligands using systems biology approaches to X-ray crystallography and chemical biology (Srinivasan *et al*, 2013), including a series of quantitative bioassays for ERα function that were statistically robust and reproducible, based on the Z'-statistic (Fig EV1A and B; see Materials and Methods). We also determined the structures of 76 distinct ERα LBD complexes bound to different ligand types, which allowed us to understand how diverse ligand scaffolds distort the active conformation of the ERα LBD. Our findings here indicate that specific structural perturbations can be tied to ligand-selective domain usage and signaling patterns, thus providing a framework for structure-based design of improved breast cancer therapeutics, and understanding the different phenotypic effects of environmental estrogens.

# Results

### Strength of AF-1 signaling does not determine cell-specific signaling

To compare ERα signaling induced by diverse ligand types, we synthesized and assayed a library of 241 ERα ligands containing 19 distinct molecular scaffolds. These include 15 *indirect modulator* series, which lack a SERM-like side chain and modulate coactivator binding indirectly from the ligand-binding pocket (Fig 2A–E; Dataset EV1) (Zheng *et al*, 2012) (Zhu *et al*, 2012) (Muthyala *et al*, 2003; Seo *et al*, 2006) (Srinivasan *et al*, 2013) (Wang *et al*, 2012) (Liao *et al*, 2014) (Min *et al*, 2013). We also generated four *direct modulator* series with side chains designed to directly dislocate h12 and thereby completely occlude the AF-2 surface (Fig 2C and E; Dataset EV1) (Kieser *et al*, 2010). Ligand profiling using our quantitative bioassays revealed a wide range of ligand-induced *GREB1* expression, reporter gene activities, ERα-coactivator interactions, and proliferative effects on MCF-7 breast cancer cells (Figs EV1 and EV2A–J). This wide variance enabled us to probe specific features of ERα signaling using ligand class analyses, and identify signaling patterns shared by specific ligand series or scaffolds.

We first asked whether direct modulation of the receptor with an extended side chain is required for cell-specific signaling. To this end, we compared the average ligand-induced *GREB1* mRNA levels in MCF-7 cells and 3×ERE-Luc reporter gene activity in Ishikawa endometrial cancer cells (E-Luc) or in HepG2 cells transfected with wild-type ERα (L-Luc ERα-WT) (Figs 3A and EV2A–C). Direct modulators showed significant differences in average activity between cell types except OBHS-ASC analogs, which had similar low agonist activities in the three cell types. The other direct modulators had low agonist activity in Ishikawa cells, no or inverse agonist activity in MCF-7 cells, and more variable activity in HepG2

liver cells. While it was known that direct modulators such as tamoxifen drive cell-specific signaling, these experiments reveal that indirect modulators also drive cell-specific signaling, since eight of fourteen classes showed significant differences in average activity (Figs 3A and EV2A–C).

Tamoxifen depends on AF-1 for its cell-specific activity (Sakamoto *et al*, 2002); therefore, we asked whether cell-specific signaling observed here is due to a similar dependence on AF-1 for activity (Fig EV1). To test this idea, we compared the average L-Luc activities of each scaffold in HepG2 cells co-transfected with wild-type ERα or with ERα lacking the AB domain (Figs 1B and EV1). While E2 showed similar L-Luc ERα-WT and ERα-ΔAB activities, tamoxifen showed complete loss of activity without the AB domain (Fig EV1B). Deletion of the AB domain significantly reduced the average L-Luc activities of 14 scaffolds (Student's *t*-test, $P \leq 0.05$) (Fig 3B). These "AF-1-sensitive" activities were exhibited by both direct and indirect modulators, and were not limited to scaffolds that showed cell-specific signaling (Fig 3A and B). Thus, the strength of AF-1 signaling does not determine cell-specific signaling.

### Identifying cell-specific signaling clusters in ERα ligand classes

As another approach to identifying cell-specific signaling, we determined the degree of correlation between ligand-induced activities in the different cell types. Here, we compared ligands within each class (Fig 3C), instead of comparing average activities (Fig 3A and B). For each ligand class or scaffold, we calculated the Pearson's correlation coefficient, *r*, for pairwise comparison of activity profiles in breast (*GREB1*), liver (L-Luc), and endometrial cells (E-Luc). The value of *r* ranges from −1 to 1, and it defines the extent to which the data fit a straight line when compounds show similar agonist/antagonist activity profiles between cell types (Fig EV3A). We also calculated the coefficient of determination, $r^2$, which describes the percentage of variance in a dependent variable such as proliferation that can be predicted by an independent variable such as *GREB1* expression. We present both calculations as $r^2$ to readily compare signaling specificities using a heat map on which the red–yellow palette indicates significant positive correlations ($P \leq 0.05$, *F*-test for nonzero slope), while the blue palette denotes negative correlations (Fig 3C–F).

This analysis revealed diverse signaling specificities that we grouped into three clusters. Scaffolds in *cluster 1* exhibited strongly correlated *GREB1* levels, E-Luc and L-Luc activity profiles across the three cell types (Fig 3C lanes 1–4), suggesting these ligands use similar ERα signaling pathways in the breast, endometrial, and liver cell types. This cluster includes WAY-C, OBHS, OBHS-N, and triarylethylene analogs, all of which are indirect modulators. *Cluster 2* contains scaffolds with activities that were positively correlated in only two of the three cell types, indicating cell-specific signaling (Fig 3C lanes 5–12). This cluster includes two classes of direct modulators (cyclofenil-ASC and WAY dimer), and six classes of indirect modulators (2,5-DTP, 3,4-DTP, S-OBHS-2 and S-OBHS-3, furan, and WAY-D). In this cluster, the correlated activities varied by scaffold. For example, 3,4-DTP, furan, and S-OBHS-2 drove positively correlated *GREB1* levels and E-Luc but not L-Luc ERα-WT activity (Fig 3C lanes 5–7). In contrast, WAY dimer and WAY-D analogs drove positively correlated *GREB1* levels and L-Luc ERα-WT but not E-Luc activity (Fig 3C lanes 8 and 9). The last set of scaffolds, *cluster 3*, displayed cell-specific activities that were not correlated in

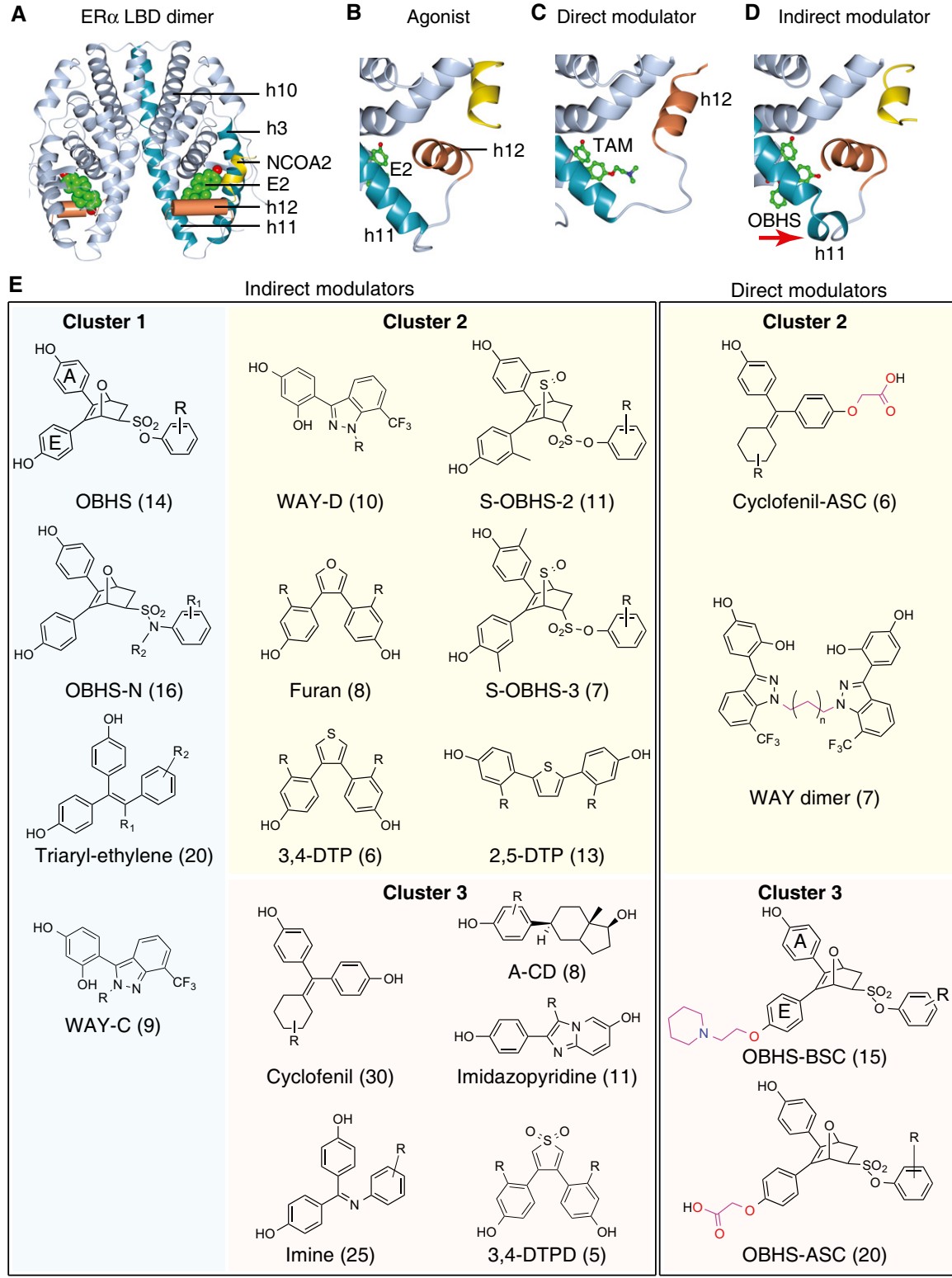

**Figure 2. Classes of compounds in the ERα ligand library.**

A    Structure of the E2-bound ERα LBD in complex with an NCOA2 peptide of (PDB 1GWR).

B–D    Structural details of the ERα LBD bound to the indicated ligands. Unlike E2 (PDB 1GWR), TAM is a direct modulator with a BSC that dislocates h12 to block the NCOA2-binding site (PDB 3ERT). OBHS is an indirect modulator that dislocates the h11 C-terminus to destabilize the h11–h12 interface (PDB 4ZN9).

E    The ERα ligand library contains 241 ligands representing 15 indirect modulator scaffolds, plus 4 direct modulator scaffolds. The number of compounds per scaffold is shown in parentheses (see Dataset EV1 for individual compound information and Appendix Supplementary Methods for synthetic protocols).

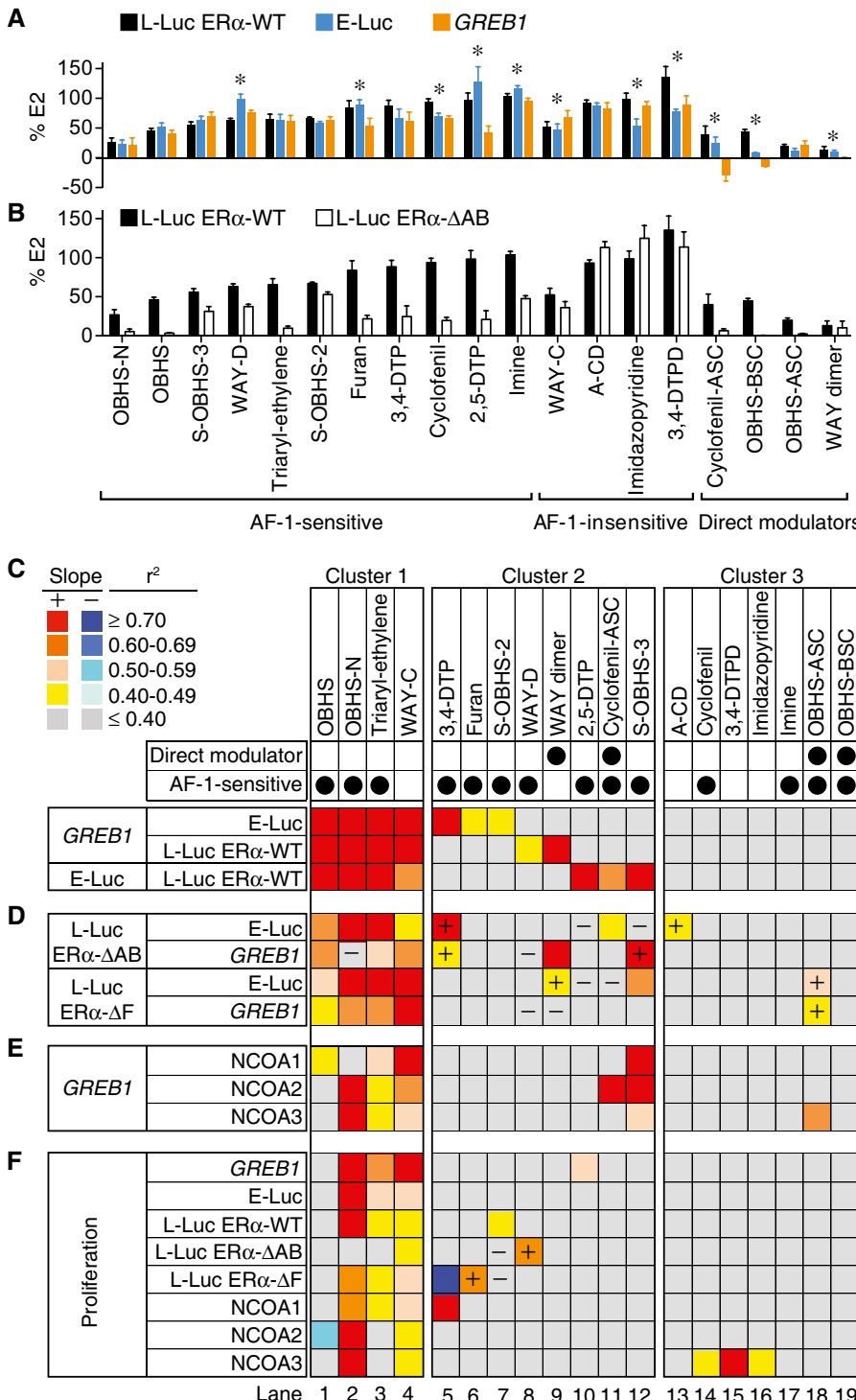

**Figure 3. Ligand-specific signaling underlies ERα-mediated cell proliferation.**

A, B  (A) Ligand-specific ERα activities in HepG2, Ishikawa and MCF-7 cells. The ligand-induced L-Luc ERα-WT and E-Luc activities and *GREB1* mRNA levels are shown by scaffold (mean + SD). (B) Ligand class analysis of the L-Luc ERα-WT and ERα-ΔAB activities in HepG2 cells. Significant sensitivity to AB domain deletion was determined by Student's *t*-test (n = number of ligands per scaffold in Fig 2). The average activities of ligands classes are shown (mean + SEM).

C–F  Correlation and regression analyses in a large test set. The $r^2$ values are plotted as a heat map. In *cluster 1*, the first three comparisons (rows) showed significant positive correlations (F-test for nonzero slope, $P \leq 0.05$). In *cluster 2*, only one of these comparisons revealed a significant positive correlation, while none was significant in *cluster 3*. +, statistically significant correlations gained by deletion of the AB or F domains. −, significant correlations lost upon deletion of AB or F domains.

Source data are available online for this figure.

any of the three cell types (Fig 3C lanes 13–19). This cluster includes two direct modulator scaffolds (OBHS-ASC and OBHS-BSC), and five indirect modulator scaffolds (A-CD, cyclofenil, 3,4-DTPD, imine, and imidazopyridine).

These results suggest that addition of an extended side chain to an ERα ligand scaffold is sufficient to induce cell-specific signaling, where the relative activity profiles of the individual ligands change between cell types. This is demonstrated by directly comparing the signaling specificities of matched OBHS (indirect modulator, cluster 1) and OBHS-BSC analogs (direct modulator, cluster 3), which differ only in the basic side chain (Fig 2E). The activities of OBHS analogs were positively correlated across the three cell types, but the side chain of OBHS-BSC analogs was sufficient to abolish these correlations (Figs 3C lanes 1 and 19, and EV3A–C).

The indirect modulator scaffolds in clusters 2 and 3 showed cell-specific signaling patterns without the extended side chain typically viewed as the primary chemical and structural mechanism driving cell-specific activity. Many of these scaffolds drove similar average activities of the ligand class in the different cell types (Fig 3A), but the individual ligands in each class had different cell-specific activities (Fig EV2A–C). Thus, examining the correlated patterns of ERα activity within each scaffold demonstrates that an extended side chain is not required for cell-specific signaling.

### Modulation of signaling specificity by AF-1

To evaluate the role of AF-1 and the F domain in ERα signaling specificity, we compared activity of truncated ERα constructs in HepG2 liver cells with endogenous ERα activity in the other cell types. The positive correlation between the L-Luc and E-Luc activities or *GREB1* levels induced by scaffolds in cluster 1 was generally retained without the AB domain, or the F domain (Fig 3D lanes 1–4). This demonstrates that the signaling specificities underlying these positive correlations are not modified by AF-1. OBHS analogs showed an average L-Luc ERα-ΔAB activity of 3.2% ± 3 (mean + SEM) relative to E2. Despite this nearly complete lack of activity, the pattern of L-Luc ERα-ΔAB activity was still highly correlated with the E-Luc activity and *GREB1* expression (Fig EV3D and E), demonstrating that very small AF-2 activities can be amplified by AF-1 to produce robust signals. Similarly, deletion of the F domain did not abolish correlations between the L-Luc and E-Luc or *GREB1* levels induced by OBHS analogs (Fig EV3F). These similar patterns of ligand activity in the wild-type and deletion mutants suggest that AF-1 and the F domain purely amplify the AF-2 activities of ligands in cluster 1.

In contrast, AF-1 was a determinant of signaling specificity for scaffolds in cluster 2. Deletion of the AB or F domain altered correlations for six of the eight scaffolds in this cluster (2,5-DTP, 3,4-DTP, S-OBHS-3, WAY-D, WAY dimer, and cyclofenil-ASC) (Fig 3D lanes 5–12). Comparing Fig 3C and D, the + and − signs indicate where the deletion mutant assays led to a gain or loss of statically significant correlation, respectively. Thus, in cluster 2, AF-1 substantially modulated the *specificity* of ligands with cell-specific activity (Fig 3D lanes 5–12). For ligands in cluster 3, we could not eliminate a role for AF-1 in determining signaling specificity, since this cluster lacked positively correlated activity profiles (Fig 3C), and deletion of the AB or F domain rarely induced such correlations (Fig 3D), except for A-CD and OBHS-ASC analogs, where deletion of the AB domain or F domain led to positive correlations with E-Luc activity

and/or *GREB1* levels (Fig 3D lanes 13 and 18). Thus, ligands in cluster 2 rely on AF-1 for both activity (Fig 3B) and signaling *specificity* (Fig 3D). As discussed below, this cell specificity derives from alternate coactivator preferences.

### Ligand-specific control of *GREB1* expression

To determine whether ligand classes control expression of native ERα target genes through the canonical linear signaling pathway, we performed pairwise linear regression analyses using ERα–NCOA1/2/3 interactions in M2H assay as independent predictors of *GREB1* expression (the dependent variable) (Figs EV1 and EV2A, F–H). In cluster 1, the recruitment of NCOA1 and NCOA2 was highest for WAY-C, followed by triaryl-ethylene, OBHS-N, and OBHS series, while for NCOA3, OBHS-N compounds induced the most recruitment and OBHS ligands were inverse agonists (Fig EV2F–H). The average induction of *GREB1* by cluster 1 ligands showed greater variance, with a range between ~25 and ~75% for OBHS and a range from full agonist to inverse agonist for the others in cluster 1 (Fig EV2A). *GREB1* levels induced by OBHS analogs were determined by recruitment of NCOA1 but not NCOA2/3 (Fig 3E lane 1), suggesting that there may be alternate or preferential use of these coactivators by different classes. However, in cluster 1, NCOA1/2/3 recruitment generally predicted *GREB1* levels (Fig 3E lanes 1–4), consistent with the canonical signaling model (Fig 1D).

For clusters 2 and 3, *GREB1* activity was generally not predicted by NCOA1/2/3 recruitment. Direct modulators showed low NCOA1/2/3 recruitment (Fig EV2F–H), but only OBHS-ASC analogs had NCOA2 recruitment profiles that predicted a full range of effects on *GREB1* levels (Figs 3E lanes 9, 11, 18–19, and EV2A). The indirect modulators in clusters 2 and 3 stimulated NCOA1/2/3 recruitment and *GREB1* expression with substantial variance (Figs 3A and EV2F–H). However, ligand-induced *GREB1* levels were generally not determined by NCOA1/2/3 recruitment (Fig 3E lanes 5–19), consistent with an alternate causality model (Fig 1E). Out of 11 indirect modulator series in cluster 2 or 3, only the S-OBHS-3 class had NCOA1/2/3 recruitment profiles that predicted *GREB1* levels (Fig 3E lane 12). These results suggest that compounds that show cell-specific signaling do not activate *GREB1*, or use coactivators other than NCOA1/2/3 to control *GREB1* expression (Fig 1E).

### Ligand-specific control of cell proliferation

To determine mechanisms for ligand-dependent control of breast cancer cell proliferation, we performed linear regression analyses across the 19 scaffolds using MCF-7 cell proliferation as the dependent variable, and the other activities as independent variables (Fig 3F). In cluster 1, E-Luc and L-Luc activities, NCOA1/2/3 recruitment, and *GREB1* levels generally predicted the proliferative response (Fig 3F lanes 2–4). With the OBHS-N compounds, NCOA3 and *GREB1* showed near perfect prediction of proliferation (Fig EV3G), with unexplained variance similar to the noise in the assays. The lack of significant predictors for OBHS analogs (Fig 3F lane 1) reflects their small range of proliferative effects on MCF-7 cells (Fig EV2I). The significant correlations with *GREB1* expression and NCOA1/2/3 recruitment observed in this cluster are consistent with the canonical signaling model (Fig 1D), where NCOA1/2/3 recruitment determines *GREB1* expression, which then drives proliferation.

Ligands in cluster 2 and cluster 3 showed a wide range of proliferative effects on MCF-7 cells (Fig EV2I). Despite this phenotypic variance, proliferation was not generally predicted by correlated NCOA1/2/3 recruitment and *GREB1* induction (Figs 3F lanes 5–19, and EV3H). Out of 15 ligand series in these clusters, only 2,5-DTP analogs induced a proliferative response that was predicted by *GREB1* levels, which were not determined by NCOA1/2/3 recruitment (Fig 3E and F lane 10). 3,4-DTP, cyclofenil, 3,4-DTPD, and imidazopyridine analogs had NCOA1/3 recruitment profiles that predicted their proliferative effects, without determining *GREB1* levels (Fig 3E and F, lanes 5 and 14–16). Similarly, S-OBHS-3, cyclofenil-ASC, and OBHS-ASC had positively correlated NCOA1/2/3 recruitment and *GREB1* levels, but none of these activities determined their proliferative effects (Fig 3E and F lanes 11–12 and 18). For ligands that show cell-specific signaling, ERα-mediated recruitment of other coregulators and activation of other target genes likely determine their proliferative effects on MCF-7 cells.

### NCOA3 occupancy at *GREB1* did not predict the proliferative response

We also questioned whether promoter occupancy by coactivators is statistically robust and reproducible for ligand class analysis using a chromatin immunoprecipitation (ChIP)-based quantitative assay, and whether it has a better predictive power than the M2H assay. ERα and NCOA3 cycle on and off the *GREB1* promoter (Nwachukwu *et al*, 2014). Therefore, we first performed a time-course study, and found that E2 and the WAY-C analog, AAPII-151-4, induced recruitment of NCOA3 to the *GREB1* promoter in a temporal cycle that peaked after 45 min in MCF-7 cells (Fig 4A). At this time point, other WAY-C analogs also induced recruitment of NCOA3 at this site to varying degrees (Fig 4B). The Z′ for this assay was 0.6, showing statistical robustness (see Materials and Methods). We prepared biological replicates with different cell passage numbers and separately prepared samples, which showed $r^2$ of 0.81, demonstrating high reproducibility (Fig 4C).

The M2H assay for NCOA3 recruitment broadly correlated with the other assays, and was predictive for *GREB1* expression and cell proliferation (Fig 3E). However, the ChIP assays for WAY-C-induced recruitment of NCOA3 to the *GREB1* promoter did not correlate with any of the other WAY-C activity profiles (Fig 4D), although the positive correlation between ChIP assays and NCOA3 recruitment via M2H assay showed a trend toward significance with $r^2 = 0.36$ and $P = 0.09$ (F-test for nonzero slope). Thus, the simplified coactivator-binding assay showed much greater predictive power than the ChIP assay for ligand-specific effects on *GREB1* expression and cell proliferation.

### ERβ activity is not an independent predictor of cell-specific activity

One difference between MCF-7 breast cancer cells and Ishikawa endometrial cancer cells is the contribution of ERβ to estrogenic response, as Ishikawa cells may express ERβ (Bhat & Pezzuto, 2001). When overexpressed in MCF-7 cells, ERβ alters E2-induced expression of only a subset of ERα-target genes (Wu *et al*, 2011), raising the possibility that ligand-induced ERβ activity may contribute to E-Luc activities, and thus underlie the lack of correlation between

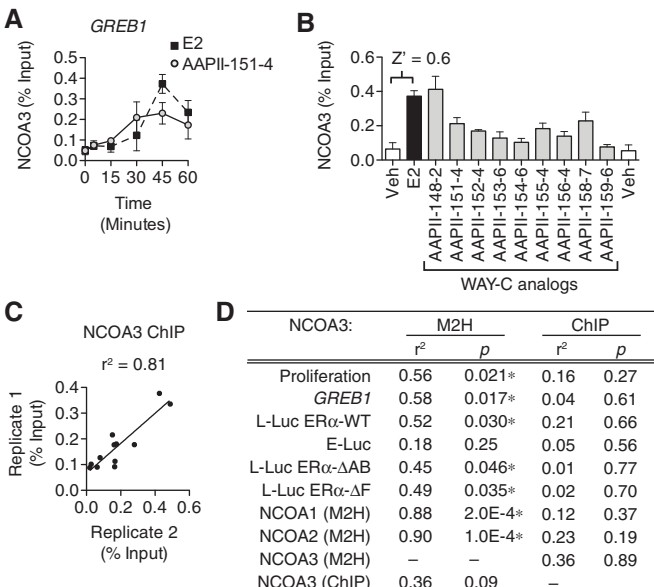

**Figure 4. NCOA3 occupancy at *GREB1* is statistically robust but does not predict transcriptional activity.**

A　Kinetic ChIP assay examining recruitment of NCOA3 to the *GREB1* gene in MCF-7 cells stimulated with E2 or the indicated WAY-C analog. The average of duplicate experiments (mean ± SEM) is shown.

B, C　NCOA3 occupancy at *GREB1* was compared by ChIP assay 45 min after stimulation with vehicle, E2, or the WAY-C analogs. In panel (B), the average recruitment of two biological replicates are shown as mean + SEM, and the Z-score is indicated. In panel (C), correlation analysis was performed for two biological replicates.

D　Linear regression analyses comparing the ability of NCOA3 recruitment, measured by ChIP or M2H, to predict other agonist activities of WAY-C analogs. *Significant positive correlation (F-test for nonzero slope, P-value).

Source data are available online for this figure.

| NCOA3: | M2H | | ChIP | |
|---|---|---|---|---|
| | $r^2$ | $p$ | $r^2$ | $p$ |
| Proliferation | 0.56 | 0.021* | 0.16 | 0.27 |
| *GREB1* | 0.58 | 0.017* | 0.04 | 0.61 |
| L-Luc ERα-WT | 0.52 | 0.030* | 0.21 | 0.66 |
| E-Luc | 0.18 | 0.25 | 0.05 | 0.56 |
| L-Luc ERα-ΔAB | 0.45 | 0.046* | 0.01 | 0.77 |
| L-Luc ERα-ΔF | 0.49 | 0.035* | 0.02 | 0.70 |
| NCOA1 (M2H) | 0.88 | 2.0E-4* | 0.12 | 0.37 |
| NCOA2 (M2H) | 0.90 | 1.0E-4* | 0.23 | 0.19 |
| NCOA3 (M2H) | – | – | 0.36 | 0.89 |
| NCOA3 (ChIP) | 0.36 | 0.09 | – | |

the E-Luc and L-Luc ERα-WT activities or *GREB1* levels induced by cell-specific modulators in cluster 2 and cluster 3 (Fig 3C).

To test this idea, we determined the L-Luc ERβ activity profiles of the ligands (Fig EV1). All direct modulator and two indirect modulator scaffolds (OBHS and S-OBHS-3) lacked ERβ agonist activity. However, the other ligands showed a range of ERβ activities (Fig EV2J). For most scaffolds, L-Luc ERβ and E-Luc activities were not correlated, except for 2,5-DTP and cyclofenil analogs, which showed moderate but significant correlations (Fig EV4A). Nevertheless, the E-Luc activities of both 2,5-DTP and cyclofenil analogs were better predicted by their L-Luc ERα-WT than L-Luc ERβ activities (Fig EV4A and B). Thus, ERβ activity was not an independent determinant of the observed activity profiles.

### Structural features of consistent signaling across cell types

To overcome barriers to crystallization of ERα LBD complexes, we developed a conformation-trapping X-ray crystallography approach using the ERα-Y537S mutation (Nettles *et al*, 2008; Bruning *et al*, 2010; Srinivasan *et al*, 2013). To further validate this approach, we solved the structure of the ERα-Y537S LBD in complex with diethylstilbestrol (DES), which bound identically in the wild-type and

ERα-Y537S LBDs, demonstrating again that this surface mutation stabilizes h12 dynamics to facilitate crystallization without changing ligand binding (Appendix Fig S1A and B) (Nettles *et al*, 2008; Bruning *et al*, 2010; Delfosse *et al*, 2012). Using this approach, we solved 76 ERα LBD structures in the active conformation and bound to ligands studied here (Appendix Fig S1C). Eleven of these structures have been published, while 65 are new, including the DES-bound ERα-Y537S LBD. We present 57 of these new structures here (Dataset EV2), while the remaining eight new structures bound to OBHS-N analogs will be published elsewhere (S. Srinivasan *et al*, in preparation). Examining many closely related structures allows us to visualize subtle structural differences, in effect using X-ray crystallography as a systems biology tool.

The indirect modulator scaffolds in cluster 1 did not show cell-specific signaling (Fig 3C), but shared common structural perturbations that we designed to modulate h12 dynamics. Based on our original OBHS structure, the OBHS, OBHS-N, and triaryl-ethylene compounds were modified with h11-directed pendant groups (Zheng *et al*, 2012; Zhu *et al*, 2012; Liao *et al*, 2014). Superposing the LBDs based on the class of bound ligands provides an ensemble view of the structural variance and clarifies what part of the ligand-binding pocket is differentially perturbed or targeted.

The 24 structures containing OBHS, OBHS-N, or triaryl-ethylene analogs showed structural diversity in the same part of the scaffolds (Figs 5A and EV5A), and the same region of the LBD—the C-terminal end of h11 (Figs 5B and C, and EV5B), which in turn nudges h12 (Fig 5C and D). We observed that the OBHS-N analogs displaced h11 along a vector away from Leu354 in a region of h3 that is unaffected by the ligands, and toward the dimer interface. For the triaryl-ethylene analogs, the displacement of h11 was in a perpendicular direction, away from Ile424 in h8 and toward h12. Remarkably, these individual inter-atomic distances showed a ligand class-specific ability to significantly predict proliferative effects (Fig 5E and F), demonstrating the feasibility of developing a minimal set of activity predictors from crystal structures.

As visualized in four LBD structures (Srinivasan *et al*, 2013), WAY-C analogs were designed with small substitutions that slightly nudge h12 Leu540, without exiting the ligand-binding pocket (Fig 5G and H). Therefore, changing h12 dynamics maintains the canonical signaling pathway defined by E2 (Fig 1D) to support AF-2-driven signaling and recruit NCOA1/2/3 for *GREB1*-stimulated proliferation.

## Ligands with cell-specific activity alter the shape of the AF-2 surface

Direct modulators like tamoxifen drive AF-1-dependent cell-specific activity by completely occluding AF-2, but it is not known how indirect modulators produce cell-specific ERα activity. Therefore, we examined another 50 LBD structures containing ligands in clusters 2 and 3. These structures demonstrated that cell-specific activity derived from altering the shape of the AF-2 surface without an extended side chain.

Ligands in cluster 2 and cluster 3 showed conformational heterogeneity in parts of the scaffold that were directed toward multiple regions of the receptor including h3, h8, h11, h12, and/or the β-sheets (Fig EV5C–G). For instance, S-OBHS-2 and S-OBHS-3 analogs (Fig 2) had similar ERα activity profiles in the different cell types (Fig EV2A–C), but the 2- versus 3-methyl substituted phenol

rings altered the correlated signaling patterns in different cell types (Fig 3B lanes 7 and 12). Structurally, the 2- versus 3-methyl substitutions changed the binding position of the A- and E-ring phenols by 1.0 Å and 2.2 Å, respectively (Fig EV5C). This difference in ligand positioning altered the AF-2 surface via a shift in the N-terminus of h12, which directly contacts the coactivator. This effect is evident in a single structure due to its 1 Å magnitude (Fig 6A and B). The shifts in h12 residues Asp538 and Leu539 led to rotation of the coactivator peptide (Fig 6C). Thus, cell-specific activity can stem from perturbation of the AF-2 surface without an extended side chain, which presumably alters the receptor–coregulator interaction profile.

The 2,5-DTP analogs showed perturbation of h11, as well as h3, which forms part of the AF-2 surface. These compounds bind the LBD in an unusual fashion because they have a phenol-to-phenol length of ~12 Å, which is longer than steroids and other prototypical ERα agonists that are ~10 Å in length. One phenol pushed further toward h3 (Fig 6D), while the other phenol pushed toward the C-terminus of h11 to a greater extent than A-CD-ring estrogens (Nwachukwu *et al*, 2014), which are close structural analogs of E2 that lack a B-ring (Fig 2). To quantify this difference, we compared the distance between α-carbons at h3 Thr347 and h11 Leu525 in the set of structures containing 2,5-DTP analogs ($n = 3$) or A-CD-ring analogs ($n = 5$) (Fig 6E). We observed a difference of 0.4 Å that was significant (two-tailed Student's *t*-test, $P = 0.002$) due to the very tight clustering of the 2,5-DTP-induced LBD conformation. The shifts in h3 suggest these compounds are positioned to alter coregulator preferences.

The 2,5-DTP and 3,4-DTP scaffolds are isomeric, but with aryl groups at obtuse and acute angles, respectively (Fig 2). The crystal structure of ERα in complex with a 3,4-DTP is unknown; however, we solved two crystal structures of ERα bound to 3,4-DTPD analogs and one structure containing a furan ligand—all of which have a 3,4-diaryl configuration (Fig 2; Datasets EV1 and EV2). In these structures, the A-ring mimetic of the 3,4-DTPD scaffold bound h3 Glu353 as expected, but the other phenol wrapped around h3 to form a hydrogen bond with Thr347, indicating a change in binding epitopes in the ERα ligand-binding pocket (Fig 6F). The 3,4-DTPD analogs also induced a shift in h3 positioning, which translated again into a shift in the bound coactivator peptide (Fig 6F). Therefore, these indirect modulators, including S-OBHS-2, S-OBHS-3, 2,5-DTP, and 3,4-DTPD analogs—all of which show cell-specific activity profiles—induced shifts in h3 and h12 that were transmitted to the coactivator peptide via an altered AF-2 surface.

To test whether the AF-2 surface shows changes in shape in solution, we used the microarray assay for real-time coregulator–nuclear receptor interaction (MARCoNI) analysis (Aarts *et al*, 2013). Here, the ligand-dependent interactions of the ERα LBD with over 150 distinct LxxLL motif peptides were assayed to define structural fingerprints for the AF-2 surface, in a manner similar to the use of phage display peptides as structural probes (Connor *et al*, 2001). Despite the similar average activities of these ligand classes (Fig 3A and B), 2,5-DTP and 3,4-DTP analogs displayed remarkably different peptide recruitment patterns (Fig 6H), consistent with the structural analyses.

Hierarchical clustering revealed that many of the 2,5-DTP analogs recapitulated most of the peptide recruitment and dismissal patterns observed with E2 (Fig 6H). However, there was a unique cluster of peptides that were recruited by E2 but not the 2,5-DTP analogs. In contrast, 3,4-DTP analogs dismissed most of the peptides

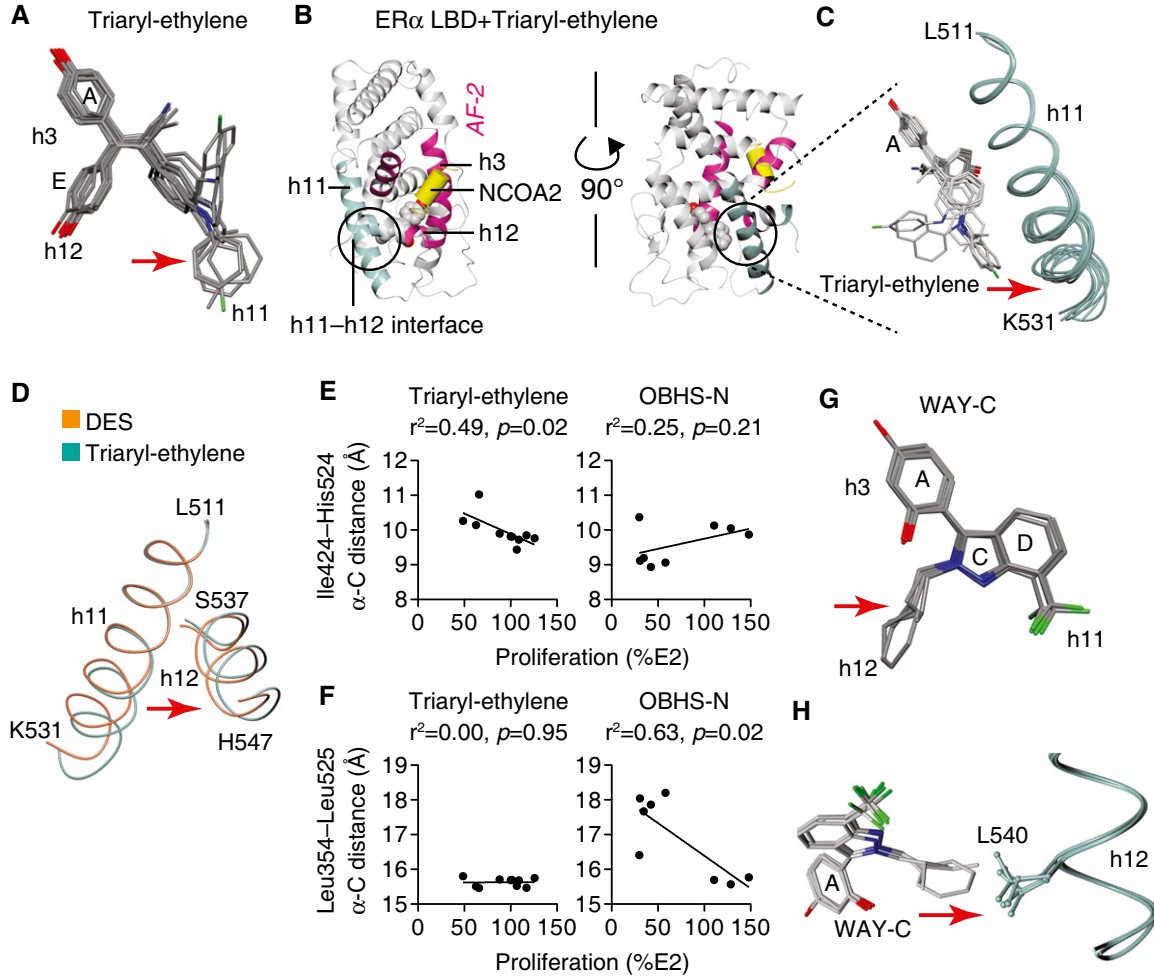

**Figure 5. Structural determinants of consistent signaling.**

A    Structure-class analysis of triaryl-ethylene analogs. Triaryl-ethylene analogs bound to the superposed crystal structures of the ERα LBD are shown. Arrows indicate chemical variance in the orientation of the different h11-directed ligand side groups (PDB 5DK9, 5DKB, 5DKE, 5DKG, 5DKS, 5DL4, 5DLR, 5DMC, 5DMF and 5DP0).

B, C   Triaryl-ethylene analogs induce variance of ERα conformations at the C-terminal region of h11. Panel (B) shows the crystal structure of a triaryl-ethylene analog-bound ERα LBD (PDB 5DLR). The h11–h12 interface (circled) includes the C-terminal part of h11. This region was expanded in panel (C), where the 10 triaryl-ethylene analog-bound ERα LBD structures (see Datasets EV1 and EV2) were superposed to show variations in the h11 C-terminus (PDB 5DK9, 5DKB, 5DKE, 5DKG, 5DKS, 5DL4, 5DLR, 5DMC, 5DMF, and 5DP0).

D    ERα LBDs in complex with diethylstilbestrol (DES) or a triaryl-ethylene analog were superposed to show that the ligand-induced difference in h11 conformation is transmitted to the C-terminus of h12 (PDB 4ZN7, 5DMC).

E, F   Inter-atomic distances predict the proliferative effects of specific ligand series. Ile424–His524 distance measured in the crystal structures correlates with the proliferative effect of triaryl-ethylene analogs in MCF-7 cells. In contrast, the Leu354–Leu525 distance correlates with the proliferative effects of OBHS-N analogs in MCF-7 cells.

G, H   Structure-class analysis of WAY-C analogs. WAY-C side groups subtly nudge h12 Leu540. ERα LBD structures bound to 4 distinct WAY-C analogs were superposed (PDB 4 IU7, 4IV4, 4IVW, 4IW6) (see Datasets EV1 and EV2).

Source data are available online for this figure.

from the AF-2 surface (Fig 6H). Thus, the isomeric attachment of diaryl groups to the thiophene core changed the AF-2 surface from inside the ligand-binding pocket, as predicted by the crystal structures. Together, these findings suggest that without an extended side chain, cell-specific activity stems from different coregulator recruitment profiles, due to unique ligand-induced conformations of the AF-2 surface, in addition to differential usage of AF-1. Indirect modulators in cluster 1 avoid this by perturbing the h11–h12 interface, and modulating the dynamics of h12 without changing the shape of AF-2 when stabilized.

# Discussion

Our goal was to identify a minimal set of predictors that would link specific structural perturbations to ERα signaling pathways that control cell-specific signaling and proliferation. We found a very strong set of predictors, where ligands in cluster 1, defined by similar signaling across cell types, showed indirect modulation of h12 dynamics via the h11–12 interface or slight contact with h12. This perturbation determined proliferation that correlated strongly with AF-2 activity, recruitment of NCOA1/2/3 family

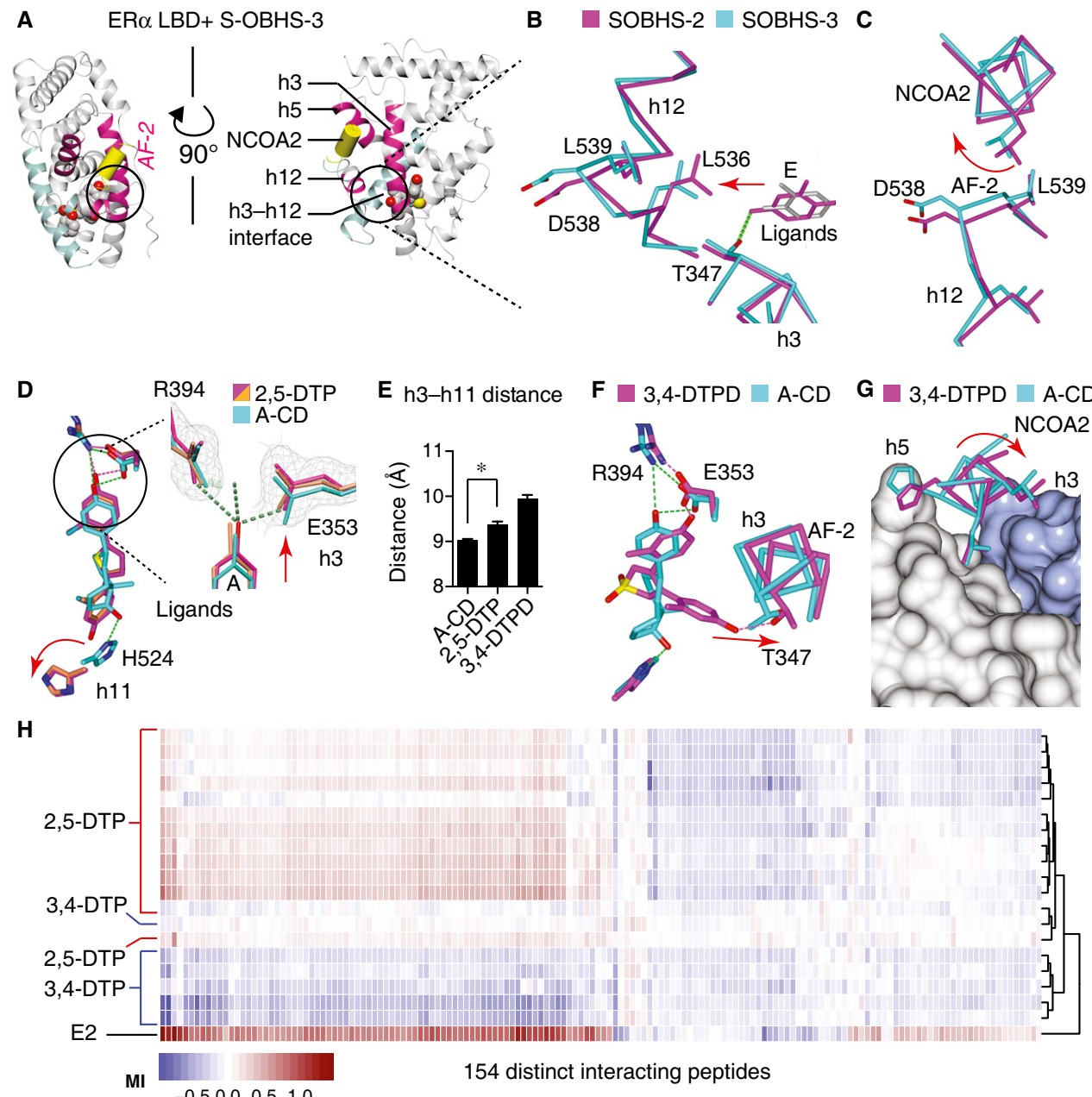

**Figure 6.   Structural correlates of cell-specific signaling.**

A–C   S-OBHS-2/3 analogs subtly distort the AF-2 surface. Panel (A) shows the crystal structure of an S-OBHS-3-bound ERα LBD (PDB 5DUH). The h3–h12 interface (circled) at AF-2 (pink) was expanded in panels (B, C). The S-OBHS-2/3-bound ERα LBDs were superposed to show shifts in h3 (panel B) and the NCOA2 peptide docked at the AF-2 surface (panel C).

D   Crystal structures show that 2,5-DTP analogs shift h3 and h11 further apart compared to an A-CD-ring estrogen (PDB 4PPS, 5DRM, 5DRJ). The $2F_o$-$F_c$ electron density map and $F_o$-$F_c$ difference map of a 2,5-DTP-bound structure (PDB 5DRJ) were contoured at 1.0 sigma and ± 3.0 sigma, respectively.

E   Average (mean + SEM) α-carbon distance measured from h3 Thr347 to h11 Leu525 of A-CD-, 2,5-DTP-, and 3,4-DTPD-bound ERα LBDs. *Two-tailed Student's t-test, $P$ = 0.002 (PDB A-CD: 5DI7, 5DID, 5DIE, 5DIG, and 4PPS; 2,5-DTP: 4IWC, 5DRM, and 5DRJ; 3,4-DTPD: 5DTV and 5DU5).

F, G   Crystal structures show that a 3,4-DTPD analog shifts h3 (F) and the NCOA2 (G) peptide compared to an A-CD-ring estrogen (PDB 4PPS, 5DTV).

H   Hierarchical clustering of ligand-specific binding of 154 interacting peptides to the ERα LBD was performed in triplicate by MARCoNI analysis.

Source data are available online for this figure.

members, and induction of the *GREB1* gene, consistent with the canonical ERα signaling pathway (Fig 1D). For ligands in cluster 1, deletion of AF-1 reduced activity to varying degrees, but did not change the underlying signaling patterns established through AF-2. In contrast, an extended side chain designed to directly reposition h12 and completely disrupt the AF-2 surface results in

cell-specific signaling. This was demonstrated with direct modulators in clusters 2 and 3. Cluster 2 was defined by ligand classes that showed correlated activities in two of the three cell types tested, while ligand classes in cluster 3 did not show correlated activities among any of the three cell types. Compared to cluster 1, the structural rules are less clear in clusters 2 and 3, but a number of indirect modulator classes perturbed the LBD conformation at the intersection of h3, the h12 N-terminus, and the AF-2 surface. Ligands in these classes altered the shape of AF-2 to affect coregulator preferences. For direct and indirect modulators in cluster 2 or 3, the canonical ERα signaling pathway involving recruitment of NCOA1/2/3 and induction of *GREB1* did not generally predict their proliferative effects, indicating an alternate causal model (Fig 1E).

These principles outlined above provide a structural basis for how the ligand–receptor interface leads to different signaling specificities through AF-1 and AF-2. It is noteworthy that regulation of h12 dynamics indirectly through h11 can virtually abolish AF-2 activity, and yet still drive robust transcriptional activity through AF-1, as demonstrated with the OBHS series. This finding can be explained by the fact that NCOA1/2/3 contain distinct binding sites for interaction with AF-1 and AF-2 (McInerney *et al*, 1996; Webb *et al*, 1998), which allows ligands to nucleate ERα–NCOA1/2/3 interaction through AF-2, and reinforce this interaction with additional binding to AF-1. Completely blocking AF-2 with an extended side chain or altering the shape of AF-2 changes the preference away from NCOA1/2/3 for determining *GREB1* levels and proliferation of breast cancer cells. AF-2 blockade also allows AF-1 to function independently, which is important since AF-1 drives tissue-selective effects *in vivo*. This was demonstrated with AF-1 knockout mice that show E2-dependent vascular protection, but not uterine proliferation, thus highlighting the role of AF-1 in tissue-selective or cell-specific signaling (Billon-Gales *et al*, 2009; Abot *et al*, 2013).

One current limitation to our approach is the identification of statistical variables that predict ligand-specific activity. Here, we examined many LBD structures and tested several variables that were not predictive, including ERβ activity, the strength of AF-1 signaling, and NCOA3 occupancy at the *GREB1* gene. Similarly, we visualized structures to identify patterns. There are many systems biology approaches that could contribute to the unbiased identification of predictive variables for statistical modeling. For example, phage display was used to identify the androgen receptor interactome, which was cloned into an M2H library and used to identify clusters of ligand-selective interactions (Norris *et al*, 2009). Also, we have used siRNA screening to identify a number of coregulators required for ERα-mediated repression of the *IL-6* gene (Nwachukwu *et al*, 2014). However, the use of larger datasets to identify such predictor variables has its own limitations, one of the major ones being the probability of false positives from multiple hypothesis testing. If we calculated inter-atomic distance matrices containing 4,000 atoms per structure × 76 ligand–receptor complexes, we would have $3 \times 10^5$ predictions. One way to address this issue is to use the cross-validation concept, where hypotheses are generated on training sets of ligands and tested with another set of ligands.

Based on this work, we propose several testable hypotheses for drug discovery. We have identified atomic vectors for the OBHS-N and triaryl-ethylene classes that predict ligand response (Fig 5E

and F). These ligands in cluster 1 drive consistent, canonical signaling across cell types, which is desirable for generating full antagonists. Indeed, the most anti-proliferative compound in the OBHS-N series had a fulvestrant-like profile across a battery of assays (S. Srinivasan *et al*, in preparation). Secondly, our finding that WAY-C compounds do not rely of AF-1 for signaling efficacy may derive from the slight contacts with h12 observed in crystal structures (Figs 3B and 5H), unlike other compounds in cluster 1 that dislocate h11 and rely on AF-1 for signaling efficacy (Figs 3B and 5C, and EV5B). Thirdly, we found ligands that achieved cell-specific activity without a prototypical extended side chain. Some of these ligands altered the shape of the AF-2 surface by perturbing the h3–h12 interface, thus providing a route to new SERM-like activity profiles by combining indirect and direct modulation of receptor structure. Incorporation of statistical approaches to understand relationships between structure and signaling variables moves us toward predictive models for complex ERα-mediated responses such as *in vivo* uterine proliferation or tumor growth, and more generally toward structure-based design for other allosteric drug targets including GPCRs and other nuclear receptors.

# Materials and Methods

### Statistical analysis

Correlation and linear regression analyses were performed using GraphPad Prism software. For correlation analysis, the degree to which two datasets vary together was calculated with the Pearson correlation coefficient ($r$). However, we reported $r^2$ rather than $r$, to facilitate comparison with the linear regression results for which we calculated and reported $r^2$ (Fig 3C–F). Significance for $r^2$ was determined using the *F*-test for nonzero slope. High-throughput assays were considered statistically robust if they show Z' > 0.5, where Z' = 1 − (3($\sigma_p$+$\sigma_n$)/|$\mu_p$−$\mu_n$|), for the mean ($\sigma$) and standard deviations ($\mu$) of the positive and negative controls (Fig EV1A and B).

### ERα ligand library

The library of compounds examined includes both previously reported (Srinivasan *et al*, 2013) and newly synthesized compound series (see Dataset EV1 for individual compound information, and Appendix Supplementary Methods for synthetic protocols).

### Luciferase reporter assays

Cells were transfected with FugeneHD reagent (Roche Applied Sciences, Indianapolis, IN) in 384-well plates. After 24 h, cells were stimulated with 10 μM compounds dispensed using a 100-nl pintool Biomeck NXP workstation (Beckman Coulter Inc.). Luciferase activity was measured 24 h later (see Appendix Supplementary Methods for more details).

### Mammalian 2-hybrid (M2H) assays

HEK293T cells were transfected with 5× UAS-luciferase reporter, and wild-type ERα-VP16 activation domain plus full-length NCOA1/

2/3-GAL4 DBD fusion protein expression plasmids, using the TransIT-LT1 transfection reagent (Mirus Bio LLC, Madison, WI). The next day, cells were stimulated with 10 μM compounds using a 100-nl pintool Biomeck NXP workstation (Beckman Coulter Inc.). Luciferase activity was measured after 24 h (see Appendix Supplementary Methods for more details).

### Cell proliferation assay

MCF-7 cells were plated on 384-well plates in phenol red-free media plus 10% FBS and stimulated with 10 μM compounds using 100-nl pintool Biomeck NXP workstation (Beckman Coulter Inc.). Cell numbers determined 1 week later (see Appendix Supplementary Methods for more details).

### Quantitative RT–PCR

MCF-7 cells were steroid-deprived and stimulated with compounds for 24 h. Total RNA was extracted and reverse-transcribed. The cDNA was analyzed using TaqMan Gene Expression Master Mix (Life Technologies, Grand Island, NY), *GREB1* and *GAPDH* (control) primers, and hybridization probes (see Appendix Supplementary Methods for more details).

### MARCoNI coregulator-interaction profiling

This assay was performed as previously described with the ERα LBD, 10 μM compounds, and a PamChiP peptide microarray (PamGene International) containing 154 unique coregulator peptides (Aarts *et al*, 2013) (see Appendix Supplementary Methods for more details).

### Protein production and X-ray crystallography

ERα protein was produced as previously described (Bruning *et al*, 2010). New ERα LBD structures (see Dataset EV2 for data collection and refinement statistics) were solved by molecular replacement using PHENIX (Adams *et al*, 2010), refined using ExCoR as previously described (Nwachukwu *et al*, 2013), and COOT (Emsley & Cowtan, 2004) for ligand-docking and rebuilding.

### Data availability

Crystal structures analyzed in this study include the following: 1GWR (Warnmark *et al*, 2002), 3ERD and 3ERT (Shiau *et al*, 1998), 4ZN9 (Zheng *et al*, 2012), 4IWC, 4 IU7, 4IV4, 4IVW, 4IW6, 4IUI, 4IV2, 4IVY and 4IW8 (Srinivasan *et al*, 2013), and 4PPS (Nwachukwu *et al*, 2014). New crystal structures analyzed in this study were deposited in the RCSB protein data bank (http://www.pdb.org): 4ZN7, 4ZNH, 4ZNS, 4ZNT, 4ZNU, 4ZNV, 4ZNW, 5DI7, 5DID, 5DIE, 5DIG, 5DK9, 5DKB, 5DKE, 5DKG, 5DKS, 5DL4, 5DLR, 5DMC, 5DMF, 5DP0, 5DRM, 5DRJ, 5DTV, 5DU5, 5DUE, 5DUG, 5DUH, 5DXK, 5DXM, 5DXP, 5DXQ, 5DXR, 5EHJ, 5DY8, 5DYB, 5DYD, 5DZ0, 5DZ1, 5DZ3, 5DZH, 5DZI, 5E0W, 5E0X, 5E14, 5E15, 5E19, 5E1C, 5DVS, 5DVV, 5DWE, 5DWG, 5DWI, 5DWJ, 5EGV, 5EI1, 5EIT.

**Expanded View** for this article is available online.

## Acknowledgements

This research received support from the National Institutes of Health (PHS 5R37DK015556 to J.A.K.; 5R33CA132022 and 5R01DK077085 to K.W.N.). S.S. is supported by the Frenchman's Creek Women for Cancer Research. The BallenIsles Men's Golf Association supports J.C.N. Research support from NSFC (81573279, 81172935, 81373255), Key Project of Ministry of Education (313040) and the Open Research Fund Program of the State Key Laboratory of Virology of China (2011002) is also gratefully acknowledged. N.J.W. was supported by the National Science Foundation (award 1359369).

X-ray diffraction data were collected at the Advanced Photon Source (APS), Argonne National Laboratory (ANL) (beamline 23-ID-B), the Stanford Synchrotron Radiation Lightsource (SSRL) (beamline 11-1), and the National Synchrotron Light Source (NSLS) at Brookhaven National Laboratory (BNL) (beamline X12C).

## Author contributions

JCN and SS contributed equally to this work. JCN and SS designed and performed experiments and wrote the manuscript; YZ, KEC, SW, JM, CD, ZL, VC, JN, NJW, JSJ, and RH performed experiments; HBZ designed experiments; and JAK and KWN designed experiments and wrote the manuscript.

## Conflict of Interest

The authors declare that they have no conflict of interest.

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
