## [Review Process File · Molecular Systems Biology]

Predictive Features of Ligand-Specific Signaling through the Estrogen Receptor

Jerome C. Nwachukwu, Sathish Srinivasan, Yangfan Zheng, Song Wang, Jian Min, Chune Dong, Zongquan Liao, Jason Nowak, Nicholas J. Wright, René Houtman, Kathryn E. Carlson, Jatinder S. Josan, Olivier Elemento, John A. Katzenellenbogen, Hai-Bing Zhou and Kendall W. Nettles

Corresponding author: Kendall W. Nettles, The Scripps Research Institute; John Katzenellenbogen, University of Illinois; Hai-Bing Zhou, Wuhan University

Review timeline:

Submission date:	17 November 2015
Editorial Decision:	20 January 2016
Revision received:	03 March 2016
Accepted:	07 March 2016

Editor: Thomas Lemberger

Transaction Report:

1st Editorial Decision

20 January 2016

Thank you again for submitting your work to Molecular Systems Biology. We have now heard back from the two referees who accepted to evaluate the study. As you will see, the referees find the topic of your study of potential interest. They raise however a series of concerns and make suggestions for modifications, which we would ask you to convincingly address in a revision of the present work. The recommendations provided by the reviewers are very clear in this regard and refer to the need of clarifications and presenting the conclusions in a more rigorous way.

 REFEREE COMMENTS

Reviewer #1:

The study brings a systematic approach towards understanding and predicting agonist/partial agonist/antagonist and selective modulator actions of estrogen receptor ligands. The authors set out to create an assay cascade to define ER ligand activities and define whether ligand actions are consistent with a linear or a branched causality model for proliferative response. Initial experiments assess effects of one compound series (OBHS ligands) +/- tamoxifen like extensions on GREB1 activation in MCF-7 cells and transfections in Ishikawa uterine cells and HepG2 liver cells. The latter cell type is strongly AF-1 dependent and responses to OBHS ligands that are not blocked by the tamoxifen-like extensions (unlike the first two cell types). Agonist activity of direct antagonists with extensions stems from pure AF-1 activity whereas indirect antagonists permit weak AF-2 activity that is amplified by AF-1. There is good correlation between ligand actions in assays that are variously weakly and strongly dependent on AF-2.

The authors then analyze a large number of compounds derived from 19 scaffolds, most of which lacked tamoxifen-like extensions. There are wide ranges of effects on GREB1 and reporter activity. For 14 scaffold series, there is strong AF-1 dependence whereas 5 are insensitive to AF-1 deletion. Comparisons of ligand effects revealed three compound clusters. Cluster 1 displayed activities that strongly correlated between assays. Cluster 2 displayed correlation between 2 of the 3 assays. Cluster 3 is not formally defined in the text but I deduce there is no correlation between assays. The latter two patterns are consistent with selective modulation. Correlations between compounds in cluster 1 are highly AF-2 dependent whereas deletion of the AB and F domain altered correlations for ligand classes in cluster 2 and one example in cluster 3. Thus, selective modulator effects are AF-1 and chemical scaffold dependent. Use of additional assays reveals correlation between GREB1 induction, MCF-7 proliferation and coactivator recruitment in mammalian two hybrid assays by cluster 1 compounds with some evidence for compound series-specific effects on coactivator preference. There is little correlation between GREB1 induction, MCF-7 proliferation and NCOA recruitment for clusters 2 and 3. Limited analysis suggests that mammalian 2 hybrid assays are better predictors of estrogenic like effects on proliferation than direct ChIP promoter occupancy assessment. Overall, results allow the authors to suggest a model of linear causality for cluster 1 compounds and branched causality for selective modulators in clusters 2 and 3.

The authors go on to analyze binding mode of ligands using X-ray structural analysis of an active conformation trapped ERalpha mutant. Cluster 1 compounds induced common structural perturbations at the base of H11, as expected. This implies that ligand-dependent control of cell proliferation is dependent on changes in H12 dynamics. Cluster 2 compounds differentially affect AF-2 surface conformation. The idea that representative cluster2 compounds differentially affect the AF-2 surface was tested by determination of ligand effects on ERalpha interactions with a peptide library. Thus, the degree of selective modulation is dependent on AF-1 and chemical scaffold effects but may relate to ligand-specific changes in the AF-2 surface.

Overall, the paper represents an impressive body of integrated work. It is obviously impossible to describe all the results in detail and the "birds eye" view that the authors have taken is entirely appropriate. This is also an exceptionally well qualified and imaginative group and I have no doubts that the experiments and analysis was carried out appropriately. In one sense, there is not much new in terms of specific results. Mechanisms of ER antagonist actions, AF-1 involvement in gene expression and synergy between AF-1 and AF-2 are very well understood. However, the correlative weight of data and ability to generate compound category wide conclusions provides new conclusions about ligand regulation of ER structure, transactivation domain usage and MCF-7 proliferation and elevates the paper, much as previous coordinated studies of activities and peptide binding profiles of a large number of androgen receptor ligands by the Donald McDonnell group clarified the continuum of ligand response.

I nevertheless have the following general suggestions and specific questions. I suspect that all of these issues could be addressed with rewrites.

1. The title promises a lot, which is not really delivered by the paper in its current format. Can the authors summarize how the data, largely functional assays, could be used to predict a phenotypic outcome for an estrogen receptor ligand? Indeed, what is a phenotypic outcome? This should be defined. I presume that the authors do not claim that they can assess ligand activity by looking at its chemical structure or its binding mode (or do they?). What then, can be predicted and how would predictions be realized? If they mean that this battery of assays will represent an integrated minimal set of tests to smoothly predict in vivo outcomes, as I suspect that they do, this should be stated clearly and explained in terms of predictive ability.

2. This is a very complicated paper, because of the large number of ligands and types and the diverse assays, but it is still sometimes hard to read and would benefit from an edit. I have included some examples. In some places, the paragraph structure is not ideal. One notable example is in the direct antagonism is sufficient for cell-specific agonist activity results section (why are there no page or line numbers?) where text is broken into separate paragraphs that do not necessarily represent discrete points. Another example is the ERbeta analysis. This issue should be introduced properly and not included in the middle of a paragraph and properly explained. As mentioned above, cluster 3 was never formally defined. There are also occasions in which the authors talk first about correlation without actually reminding us of the results of the assays (largely in supplementary

figure 3). For example, in the description of Fig 3D, did any of the compounds that did not display correlation between effects on GREB1 induction and NCOA recruitment have any effect on NCOA induction. Likewise in Fig.3E, did any compounds actually influence proliferation? While this can be ascertained from Fig. S3, it takes a great deal of thinking and consideration to understand and the reader could be helped by one or two extra sentences of explanation. It is not obvious why the fact that OBHS analogs showed only 3.2% of L-Luc ER α -delta AB activity relative to E2 is "remarkable "without a great deal of thought. The authors should spell this out in the opening sentence of the paragraph.

3. The structural studies are extensive and represent a technical tour de force, yet are also limited. It seems that the authors can verify predictions about binding mode and effects of cluster1 compounds, but cannot really make many conclusions about the way that different members of the cluster 2 DTP series might behave in specific instances. For example, members of the 3,4DTP series consistently distort AF-2 and display a tamoxifen-like peptide binding profile in the MaRCoNI assay, yet some compounds are pretty good at GREB1 induction and induction of proliferation in MCF-7 cells, presumably different from tamoxifen. It would be good to briefly discuss the differential behavior of some of these ligands with reference to the structure and explain the limitations of structural analysis in predicting specific compound activities.

4. Other limitations of the structural studies should be acknowledged. Cluster 3 compounds do not appear to be discussed. What conclusions can be drawn, if any? Am I right in thinking that there is no obvious way to explain AF-1 independence of WAY-C compounds versus other compounds from this analysis? I believe that it is also impossible to rule out differential effects on organization of other regions of the LBD. Is this true? It would also be good to know what various compounds that were crystallized with ER α Y537S actually do to activity of ER α Y537S in the context of transfection assays described here? This would help us to interpret the structural studies.

5. Finally, I am concerned about apparent discrepancies between the mammalian 2-hybrid assay (which uses NCoA1-3) and the MaRCoNI assay in Fig. 6. I do not see much obvious difference between the two DTP series in mammalian 2-hybrid assays in Fig. S3, but large differences between effects of the two series in the MaRCoNI assay, even though I believe that NCoA peptides are represented on the array. Am I correct and, if so, how can this contradiction be reconciled?

Reviewer #3:

The manuscript by Nwachukwu et al describes functional and structural results of a series of estrogen receptor ligand that specifically target the estrogen receptor alpha (ER α). The latter is a driving force of cell proliferation in a large fraction of the breast cancers. Several therapeutic strategies have been developed to treat these cancers, in particular by using ER selective modulators (SERMs), aromatase inhibitors or ER α down-regulators (SERDs) that inhibits ER signaling through receptor degradation. SERMs, such as tamoxifen, have tissue-selectivity, where they can antagonize ER α activity in breast cancer cells, but still maintain agonist signaling in other tissues.

In order to identify the molecular determinants of tissue-selective modulation and breast cancer cell proliferation, the authors used a statistical approach of the ER α signaling pathway, based on the analysis of a series of functional assays for more than 240 ER α ligands, encompassing 19 different chemical scaffolds. In addition, domain deletion analyses and crystallographic studies of 76 different ligand-bound ER α LBD structures were carried out in order to gain insight into the structural determinants of selective modulation and (as the authors say) to 'reveal the molecular underpinnings of how ligands achieve allosteric signaling outcomes'.

The goals of the study are: i) to identify the structural and molecular rules for selective modulation by ER α ligands; ii) the understanding of the structural and molecular control of selective modulation, iii) the evaluation of the different roles for AF-1 by ligand class, and iv) (as suggested in title) the prediction of phenotypic outcomes for the ER α ligands.

The goals of the study reported in the manuscript are ambitious and already partially answering one of these questions would shed light on critical issues in the field of estrogen receptor and more generally, in the whole field of nuclear receptor.

A lot of useful data has been produced in this work. Unfortunately, we feel that the data is not clearly presented or interpreted and the manuscript does not quite meet the expectations. In addition, the text of the manuscript is too lengthy making the reading tedious. Sections in the results and in the discussion that refer to already published data or which represent introductive parts should be re-shuffled and condensed to make the text more focused around the main issues and corresponding results.

Below is a list of detailed comments.

1. Most of what is described in the first part of the results describing Figure 1 and Figures S1 and S2A-D with the corresponding text (2 1/2 pages) represents an extension of the introduction part. Important information described in this part should be put in the introduction section. As a consequence the introduction should be re-written in a more focused manner.
2. In the Results part, two consecutive sections are entitled 'Direct antagonism is sufficient for cell-specific agonist activity' for the one and 'Direct antagonism is not required for cell-specific agonist activity' for the other. While the first section, based on OBHS and OBHS-BSC analogs, describes the fact that direct antagonism is sufficient to drive cell-specific activity profiles, the other one considers the large test set of ligands and refines the conclusions. The way the two parts are written in an apparently chronological fashion, maybe reflecting the way the experiments were performed. However, what counts are the main observations and conclusions. We suggest the two parts to be put together and the main conclusions drawn only once.
3. Similarly, both sections evaluate the use of AF-1 for signaling first for OBHS and OBHS-BSC analogs and then with a large test sets. Similar observations are made (e.g. 1st part: 'direct and indirect antagonists utilize AF-1 differently'; 2nd part: ER α ligand series differ widely in their use of AF-1 for signaling,). Right after these two sections comes the next one which is entitled 'Different roles for AF-1 by ligand class'. All three parts discuss the same aspect of the work. Therefore, the analysis and the discussion of the role of AF-1 should be done in a separate and unique section.
4. The reading, the analysis and the understanding would be largely facilitated if the different classifications of the ligands could be found more easily. In the paper, ligands are grouped into direct and indirect antagonists, or into AF-1 sensitive and AF-1 insensitive, or finally into clusters 1, 2, and 3 based on activity data in different cell types. It is laborious to find the three types of classification for a given ligand. In addition in Table 1, is the classification of direct versus indirect antagonists made column-wise, all the indirect antagonists being on the left and the direct ones on the right? This should be stated!
5. The authors say that surprisingly most of the scaffolds in clusters 2 and -3 lack a SERM-like side chain (=direct antagonists). If we count well out of 15 classes in total, 9 are indirect and 6 direct, less than 2/3 of the ligand classes.
6. In the part 'Selective modulators control cell proliferation via non-canonical pathway': could the authors discuss a bit more deeply what they mean with 'non-canonical'?
7. In the part 'Structural features of consistent signaling across cell types', the authors should state in the main body of the text (not only refer to Fig S5C) that out the 76 structures analysed, 56 new ones (+ 1 with DES) are included in the paper, 8 being considered in another paper and 10 already published!
8. Again in this part, out the 4 ligands in class 1, one is an indirect antagonist (OBHS), the other 3 being direct antagonists. Still the authors do not consider this aspect in their discussion focused on the modulation of H12 dynamics as the main driver of non-selective modulation. In particular, what makes the indirect antagonist OBHS (Fig 5A) not being selective? This is not clear.
9. In the part 'Structural features of selective modulation', the authors discuss the effects of altering the shape of the AF-2 surface on the selective modulation. Could the authors be more precise, what does change in the surface? Can they illustrate this for the ligands considered in the text?
10. In addition to the superimposition of A-CD and 2,5-DTP (Figs 6C, 6D), It would be nice and even more convincing to show portion of the electron density in this case discussed where subtle differences are observed between the structures.
11. Last part of the Results section: the authors should discuss what makes the indirect antagonists belonging to class 1 not being selective modulators compared to the other indirect antagonists which are selective? This is not clearly stated.
12. In the Discussion section:

- Not clear what are the characteristics of ligands that are non-selective (see also point 8.) versus selective modulators. It seems to us that predictions of outcomes from a given ligand are hard to make, especially for indirect antagonists! Authors should be more convincing in their interpretation.
 - We do not believe that crystallization is mainly hampered by lack of proper folding in bacteria (it can but not as the most frequently found motif), but rather because of polydispersity of conformational states of the protein, the presence of disordered region or the lack of structurally crucial PTMs,
 - The 3rd paragraph of this section discusses an already published strategy and does not bring anything new to the discussion of the data.
 - The authors claim that they identified from their structural analyses 'new structural rules' for how the ligand-receptor interface leads to differential domain usage. What are the rules?
 - Similarly (5th paragraph) what are the structural characteristics of antagonists that were identified to use alternate causality pathways?
- We have not been able to really understand, from the discussion of the statistical analysis of functional and structural data, what the structural rules for selective modulation and its control are. 'Prediction of phenotypic outcomes for the ER α ligands', as stipulated in the main title does not seem a goal easily within reach.

1st Revision - authors' response

03 March 2016

Reviewer #1:

The study brings a systematic approach towards understanding and predicting agonist/partial agonist/antagonist and selective modulator actions of estrogen receptor ligands. The authors set out to create an assay cascade to define ER ligand activities and define whether ligand actions are consistent with a linear or a branched causality model for proliferative response. Initial experiments assess effects of one compound series (OBHS ligands) +/- tamoxifen like extensions on GREB1 activation in MCF-7 cells and transfections in Ishikawa uterine cells and HepG2 liver cells. The latter cell type is strongly AF-1 dependent and responses to OBHS ligands that are not blocked by the tamoxifen-like extensions (unlike the first two cell types). Agonist activity of direct antagonists with extensions stems from pure AF-1 activity whereas indirect antagonists permit weak AF-2 activity that is amplified by AF-1. There is good correlation between ligand actions in assays that are variously weakly and strongly dependent on AF-2.

The authors then analyze a large number of compounds derived from 19 scaffolds, most of which lacked tamoxifen-like extensions. There are wide ranges of effects on GREB1 and reporter activity. For 14 scaffold series, there is strong AF-1 dependence whereas 5 are insensitive to AF-1 deletion. Comparisons of ligand effects revealed three compound clusters. Cluster 1 displayed activities that strongly correlated between assays. Cluster 2 displayed correlation between 2 of the 3 assays. Cluster 3 is not formally defined in the text but I deduce there is no correlation between assays. The latter two patterns are consistent with selective modulation. Correlations between compounds in cluster 1 are highly AF-2 dependent whereas deletion of the AB and F domain altered correlations for ligand classes in cluster 2 and one example in cluster 3. Thus, selective modulator effects are AF-1 and chemical scaffold dependent. Use of additional assays reveals correlation between GREB1 induction, MCF-7 proliferation and coactivator recruitment in mammalian two hybrid assays by cluster 1 compounds with some evidence for compound series-specific effects on coactivator preference. There is little correlation between GREB1 induction, MCF-7 proliferation and NCOA recruitment for clusters 2 and 3. Limited analysis suggests that mammalian 2 hybrid assays are better predictors of estrogenic like effects on proliferation than direct ChIP promoter occupancy assessment. Overall, results allow the authors to suggest a model of linear causality for cluster 1 compounds and branched causality for selective modulators in clusters 2 and 3.

The authors go on to analyze binding mode of ligands using X-ray structural analysis of an active conformation trapped ER α mutant. Cluster 1 compounds induced common structural perturbations at the base of H11, as expected. This implies that ligand-dependent control of cell proliferation is dependent on changes in H12 dynamics. Cluster 2 compounds differentially affect AF-2 surface conformation. The idea that representative cluster2 compounds differentially affect the AF-2 surface was tested by determination of ligand effects on ER α interactions with a peptide

library. Thus, the degree of selective modulation is dependent on AF-1 and chemical scaffold effects but may relate to ligand-specific changes in the AF-2 surface.

Overall, the paper represents an impressive body of integrated work. It is obviously impossible to describe all the results in detail and the "birds eye" view that the authors have taken is entirely appropriate. This is also an exceptionally well qualified and imaginative group and I have no doubts that the experiments and analysis was carried out appropriately. In one sense, there is not much new in terms of specific results. Mechanisms of ER antagonist actions, AF-1 involvement in gene expression and synergy between AF-1 and AF-2 are very well understood. However, the correlative weight of data and ability to generate compound category wide conclusions provides new conclusions about ligand regulation of ER structure, transactivation domain usage and MCF-7 proliferation and elevates the paper, much as previous coordinated studies of activities and peptide binding profiles of a large number of androgen receptor ligands by the Donald McDonnell group clarified the continuum of ligand response.

I nevertheless have the following general suggestions and specific questions. I suspect that all of these issues could be addressed with rewrites.

1. The title promises a lot, which is not really delivered by the paper in its current format.

We have provided an alternate title (page 1).

Can the authors summarize how the data, largely functional assays, could be used to predict a phenotypic outcome for an estrogen receptor ligand? Indeed, what is a phenotypic outcome? This should be defined. I presume that the authors do not claim that they can assess ligand activity by looking at its chemical structure or its binding mode (or do they?). What then, can be predicted and how would predictions be realized? If they mean that this battery of assays will represent an integrated minimal set of tests to smoothly predict in vivo outcomes, as I suspect that they do, this should be stated clearly and explained in terms of predictive ability.

Thank you for this suggestion. We added this to the introduction: "Our long term goal is to be able to predict proliferative or anti-proliferative activity of a ligand in different tissues from its crystal structure by identifying different structural perturbations that lead to specific signaling outcomes related to proliferative drive." (Page 5, line 18)

We have modified the discussion to explicitly state what we can now predict and what are the next steps, which as you said, is to identify a minimum set of tests for prediction. (Page 20, line 16)

Thank you for asking us if we can predict activity from the structures. We tested this for the two series where we had enough structures for statistical analyses, and we could identify a particular perturbation. This new analysis is on page 17 line 9 and in Figure 5E-F

2. This is a very complicated paper, because of the large number of ligands and types and the diverse assays, but it is still sometimes hard to read and would benefit from an edit. I have included some examples. In some places, the paragraph structure is not ideal. One notable example is in the direct antagonism is sufficient for cell-specific agonist activity results section (why are there no page or line numbers?) where text is broken into separate paragraphs that do not necessarily represent discrete points.

We have condensed text into more coherent paragraphs throughout the Results Section

Another example is the ERbeta analysis. This issue should be introduced properly and not included in the middle of a paragraph and properly explained

We expanded on this point in a separate section (page 15).

As mentioned above, cluster 3 was never formally defined.

We added the following to the Results section:

The last set of scaffolds, *cluster 3*, displayed cell-specific activities that were not correlated in any of the three cell types (page 9, line 16).

We added this to the discussion:

Cluster 2 was defined by ligand classes that showed correlated activities in two of the three cell types tested, while ligand classes in cluster 3 did not show correlated activities among any of the three cell types (page 21, line 4).

There are also occasions in which the authors talk first about correlation without actually reminding us of the results of the assays (largely in supplementary figure 3) For example, in the description of Fig 3D, did any of the compounds that did not display correlation between effects on GREB1 induction and NCOA recruitment have any effect on NCOA induction. Likewise in Fig.3E, did any compounds actually influence proliferation? While this can be ascertained from Fig. S3, it takes a great deal of thinking and consideration to understand and the reader could be helped by one or two extra sentences of explanation.

We have made a number of changes to address this.

We added the assay data for the cell-specific assays to Figure 3A, and added the data on deleting AF-1 in Figure 3B. We added an additional results section describing these data on Page 6 line 19. We describe the averages for NCOA1/2/3 starting on page 12, line 2. We describe the averages for *GREB1* starting on page 12, line 6. We describe the averages for proliferation on page 13 line 12-18, including a sentence stating that indirect and direct modulators had "...a wide range of proliferative effects on MCF-7 cells" (page 13, second paragraph)

It is not obvious why the fact that OBHS analogs showed only 3.2% of L-Luc ERalpha delta AB activity relative to E2 is "remarkable "without a great deal of thought. The authors should spell this out in the opening sentence of the paragraph.

Good point. We added text explaining this finding in the Results (page 10, line 20 - page 11, line 1) and Discussion sections (page 21, lines 15-18).

3. The structural studies are extensive and represent a technical tour de force, yet are also limited. It seems that the authors can verify predictions about binding mode and effects of cluster1 compounds, but cannot really make many conclusions about the way that different members of the cluster 2 DTP series might behave in specific instances. For example, members of the 3,4DTP series consistently distort AF-2 and display a tamoxifen-like peptide binding profile in the MaRCoNI assay, yet some compounds are pretty good at GREB1 induction and induction of proliferation in MCF-7 cells, presumably different from tamoxifen. It would be good to briefly discuss the differential behavior of some of these ligands with reference to the structure and explain the limitations of structural analysis in predicting specific compound activities.

We added a statement to the Discussion (page 21, line 7) highlighting our limited understanding of the structural features driving clusters 2 and 3.

We added a paragraph to the discussion to highlight the limitations of hand picking the predictive variables, starting on page 22, line 6.

4. Other limitations of the structural studies should be acknowledged. Cluster 3 compounds do not appear to be discussed.

What conclusions can be drawn, if any?

The limitation of our structural studies is discussed on page 22, line 16-21.

Cluster 3 is presented in the results (page 9, line 16), and discussed (page 21, lines 5-6).

Am I right in thinking that there is no obvious way to explain AF-1 independence of WAY-C compounds versus other compounds from this analysis?

This is a great suggestion. We added a sentence to the Discussion on page 23, line 4 on using those structures to generate new hypotheses for testing in other chemical series, that the slight contact with h12 drives AF-1 independence.

I believe that it is also impossible to rule out differential effects on organization of other regions of the LBD. Is this true?

This got us thinking that we should calculate interatomic distances and use that as a predictive variable. It worked better than we would have expected and is now in Figure 5E-F and described on page 17 line 9. This also gave us the idea that in future experiments we could calculate the full matrix of interatomic distances to probe all areas of the receptor. This is described in the discussion on page 22 line 18.

It would also be good to know what various compounds that were crystallized with ER α Y537S actually do to activity of ER α Y537S in the context of transfection assays described here? This would help us to interpret the structural studies.

We have assayed all the compounds for activity profiles on the mutant but it is not simple to interpret, especially without proliferation data on the mutant. We are in the process of introducing the mutant allele into MCF-7 cells. These data sets are probably going to produce several manuscripts and could not be adequately addressed in a paragraph here.

5. Finally, I am concerned about apparent discrepancies between the mammalian 2-hybrid assay (which uses NCoA1-3) and the MaRCoNI assay in Fig. 6. I do not see much obvious difference between the two DTP series in mammalian 2-hybrid assays in Fig. S3, but large differences between effects of the two series in the MaRCoNI assay, even though I believe that NCoA peptides are represented on the array. Am I correct and, if so, how can this contradiction be reconciled?

The paragraph starting on page 19, line 18 describes how we are using the peptide array as a structural probe of just the AF2 surface, similar to phage display with random peptides. We do not discuss this as relevant to in vivo coactivator biology.

Page 21 line 18 describes how the full length NCOAs bind AF-1 as well as AF-2. The M2H profiles of NCOAs with the LBD would more closely resemble the peptide binding assays.

Page 22 line 14 describes how we can use shRNA screening to identify the relevant coregulators.

Reviewer #3:

The manuscript by Nwachukwu et al describes functional and structural results of a series of estrogen receptor ligand that specifically target the estrogen receptor alpha (ER α). The latter is a driving force of cell proliferation in a large fraction of the breast cancers. Several therapeutic strategies have been developed to treat these cancers, in particular by using ER selective modulators (SERMs), aromatase inhibitors or ER α down-regulators (SERDs) that inhibits ER signaling through receptor degradation. SERMs, such as tamoxifen, have tissue-selectivity, where they can antagonize ER α activity in breast cancer cells, but still maintain agonist signaling in other tissues. In order to identify the molecular determinants of tissue-selective modulation and breast cancer cell proliferation, the authors used a statistical approach of the ER α signaling pathway, based on the analysis of a series of functional assays for more than 240 ER α ligands, encompassing 19 different chemical scaffolds. In addition, domain deletion analyses and crystallographic studies of 76 different ligand-bound ER α LBD structures were carried out in order to gain insight into the structural determinants of selective modulation and (as the authors say) to 'reveal the molecular underpinnings of how ligands achieve allosteric signaling outcomes'.

The goals of the study are: i) to identify the structural and molecular rules for selective

modulation by ER α ligands; ii) the understanding of the structural and molecular control of selective modulation, iii) the evaluation of the different roles for AF-1 by ligand class, and iv) (as suggested in title) the prediction of phenotypic outcomes for the ER α ligands. The goals of the study reported in the manuscript are ambitious and already partially answering one of these questions would shed light on critical issues in the field of estrogen receptor and more generally, in the whole field of nuclear receptor. A lot of useful data has been produced in this work. Unfortunately, we feel that the data is not clearly presented or interpreted and the manuscript does not quite meet the expectations. In addition, the text of the manuscript is too lengthy making the reading tedious. Sections in the results and in the discussion that refer to already published data or which represent introductory parts should be re-shuffled and condensed to make the text more focused around the main issues and corresponding results. Below is a list of detailed comments.

1. Most of what is described in the first part of the results describing Figure 1 and Figures S1 and S2A-D with the corresponding text (2 1/2 pages) represents an extension of the introduction part. Important information described in this part should be put in the introduction section. As a consequence the introduction should be re-written in a more focused manner.

We have moved these paragraphs to the Introduction, which has been revised (page 4-6)

2. In the Results part, two consecutive sections are entitled 'Direct antagonism is sufficient for cell-specific agonist activity' for the one and 'Direct antagonism is not required for cell-specific agonist activity' for the other. While the first section, based on OBHS and OBHS-BSC analogs, describes the fact that direct antagonism is sufficient to drive cell-specific activity profiles, the other one considers the large test set of ligands and refines the conclusions. The way the two parts are written in an apparently chronological fashion, maybe reflecting the way the experiments were performed. However, what counts are the main observations and conclusions. We suggest the two parts to be put together and the main conclusions drawn only once.

We have combined the sections

3. Similarly, both sections evaluate the use of AF-1 for signaling first for OBHS and OBHS-BSC analogs and then with a large test sets. Similar observations are made (e.g. 1st part: 'direct and indirect antagonists utilize AF-1 differently'; 2nd part: ER α ligand series differ widely in their use of AF-1 for signaling, ...). Right after these two sections comes the next one which is entitled 'Different roles for AF-1 by ligand class'. All three parts discuss the same aspect of the work. Therefore, the analysis and the discussion of the role of AF-1 should be done in a separate and unique section.

We combined these sections into a new one "Modulation of signaling specificity by AF-1" (page 10, line 13) that discussed all the data on the correlation analysis from Figure 3D. However, we also added a new set of analyses in Figure 3A-B and these are discussed separately on page 7, line 10.

4. The reading, the analysis and the understanding would be largely facilitated if the different classifications of the ligands could be found more easily. In the paper, ligands are grouped into direct and indirect antagonists, or into AF-1 sensitive and AF-1 insensitive, or finally into clusters 1, 2, and 3 based on activity data in different cell types. It is laborious to find the three types of classification for a given ligand. In addition in Table 1, is the classification of direct versus indirect antagonists made column-wise, all the indirect antagonists being on the left and the direct ones on the right? This should be stated!

We have made a lot of changes throughout the text to try to make it easier for the reader to keep track of the different classifications. Figure 2E now shows the ligand scaffolds and identified the direct or indirect modulators, and clusters

from Figure 3C. In figure 3C we added two rows at the top to identify the indirect/direct modulators and AF-1 dependence. In Figure EV2, which has all the raw data, we identified the clusters, and the direct/indirect modulators.

We have also changed some terminology to help clarify what we mean. We were using indirect antagonist for historical reasons, but here we are referring to classes of ligands that have a full range of activities, so we changed it to “direct/indirect modulators”. We then stopped using modulator or modulation to describe differences in signaling between cell types, which we now refer to throughout the text as cell-specific activity.

5. The authors say that surprisingly most of the scaffolds in clusters 2 and -3 lack a SERM-like side chain (=direct antagonists). If we count well out of 15 classes in total, 9 are indirect and 6 direct, less than 2/3 of the ligand classes.

See response to comment 4

6. In the part 'Selective modulators control cell proliferation via non-canonical pathway': could the authors discuss a bit more deeply what they mean with 'non-canonical'?

We have modified the Introduction and Discussion to more clearly identify the conceptual framework in terms of identifying predictive variables. See page 5 line 10 through page 6 line 6.

7. In the part 'Structural features of consistent signaling across cell types', the authors should state in the main body of the text (not only refer to Fig S5C) that out of the 76 structures analysed, 56 new ones (+ 1 with DES) are included in the paper, 8 being considered in another paper and 10 already published!

We have stated this in the main text (page 16, lines 10-15)

8. Again in this part, out of the 4 ligands in class 1, one is an indirect antagonist (OBHS), the other 3 being direct antagonists. Still the authors do not consider this aspect in their discussion focused on the modulation of H12 dynamics as the main driver of nonselective modulation. In particular, what makes the indirect antagonist OBHS (Fig 5A) not being selective? This is not clear.

See comment 4 above about clarifying the classifications. You are right; it was confusing in the original.

9. In the part 'Structural features of selective modulation', the authors discuss the effects of altering the shape of the AF-2 surface on the selective modulation. Could the authors be more precise, what does change in the surface? Can they illustrate this for the ligands considered in the text?

We show in Figure 6A-C how changes in the shape of the h12-N-terminus portion of the AF-2 surface are transmitted to the coactivator peptide. In Figure 6F-G we show how rotation of the h3 portion of the AF2 surface does the same thing.

10. In addition to the superimposition of A-CD and 2,5-DTP (Figs 6C, 6D), It would be nice and even more convincing to show portion of the electron density in this case discussed where subtle differences are observed between the structures.

Good idea. We have revised this Figure to show the electron density of 2,5-DTP bound side chains that typically form H-bonds with the A-rings of ER ligands (Fig. 6D).

11. Last part of the Results section: the authors should discuss what makes the indirect antagonists belonging to class 1 not being selective modulators compared to the other indirect antagonists which are selective? This is not clearly stated.

See response to comment 4. We also added the sentence “Indirect modulators in cluster 1 avoid this by perturbing the h11–h12 interface, and modulating the dynamics of h12 without changing the shape of AF-2 when stabilized.” (page 20, line 12).

12. In the Discussion section:

- Not clear what are the characteristics of ligands that are non-selective (see also point 8.) versus selective modulators.

See response to comment 4

It seems to us that predictions of outcomes from a given ligand are hard to make, especially for indirect antagonists! Authors should be more convincing in their interpretation.

We added some of the raw data to Figure EV3G to better illustrate the conclusions of Figure 3. We also added Figure 5E-F to illustrate the predictive power of the structures for some ligands. We rewrote much of the Discussion (page 20) to better highlight what we can and cannot do with the current data.

- We do not believe that crystallization is mainly hampered by lack of proper folding in bacteria (it can but not as the most frequently found motif), but rather because of polydispersity of conformational states of the protein, the presence of disordered region or the lack of structurally crucial PTMs,
- The 3rd paragraph of this section discusses an already published strategy and does not bring anything new to the discussion of the data.

We removed this paragraph and replaced it with a sentence on page 16 line 3 that points to the relevant references.

- The authors claim that they identified from their structural analyses 'new structural rules' for how the ligand-receptor interface leads to differential domain usage. What are the rules?
- Similarly (5th paragraph) what are the structural characteristics of antagonists that were identified to use alternate causality pathways?
We have not been able to really understand, from the discussion of the statistical analysis of functional and structural data, what the structural rules for selective modulation and its control are.

We rewrote the Discussion (page 20) to more clearly state what rules we learned, and how this could be used in new drug discovery.

'Prediction of phenotypic outcomes for the ERα ligands', as stipulated in the main title does not seem a goal easily within reach.

We have provided an alternative title (page 1). We hope the new data in Figure EV3G and Figure 5E-F are more convincing.